# Risk of Developing Metabolic Syndrome Is Affected by Length of Daily Siesta: Results from a Prospective Cohort Study

**DOI:** 10.3390/nu13114182

**Published:** 2021-11-22

**Authors:** Anne Katherine Gribble, Carmen Sayón-Orea, Maira Bes-Rastrollo, Stefanos N. Kales, Ryutaro Shirahama, Miguel Ángel Martínez-González, Alejandro Fernandez-Montero

**Affiliations:** 1School of Medicine, University of Sydney, Camperdown 2050, Australia; agri5188@uni.sydney.edu.au; 2Department of Preventive Medicine and Public Health, School of Medicine, University of Navarra, 31008 Pamplona, Spain; msayon@unav.es (C.S.-O.); mbes@unav.es (M.B.-R.); mamartinez@unav.es (M.Á.M.-G.); 3Illawarra and Shoalhaven Local Health District, NSW Health, Wollongong 2500, Australia; 4Navarra Institute for Health Research (IdiSNa), 31008 Pamplona, Spain; 5Navarra Public Health Institute, 31003 Pamplona, Spain; 6CIBERobn, Instituto de Salud Carlos III, 28029 Madrid, Spain; 7Department of Environmental Health, Harvard T.H Chan School of Public Health, Boston, MA 02115, USA; stefokali@aol.com (S.N.K.); md.shirahama@live.jp (R.S.); 8Department of Occupational Medicine, Cambridge Health Alliance, Harvard Medical School, Cambridge, MA 02139, USA; 9Department of Public Health, Graduate School of Medicine, Juntendo University, Tokyo 113-8421, Japan; 10Faculty of Science and Technology, Keio University, Kanagawa 223-8522, Japan; 11Department of Nutrition, Harvard T.H Chan School of Public Health, Boston, MA 02115, USA; 12Department of Occupational Medicine, University of Navarra Clinic, Av. Pio XII, 36, 31008 Pamplona, Spain

**Keywords:** siesta, nap, metabolic syndrome, obesity, dyslipidaemia, hypertension

## Abstract

Background: Siesta has been associated with increased incidence of cardiovascular disease but the mechanism remains unclear. New studies into the relationship between siesta and metabolic syndrome have identified siesta length as a crucial differential, suggesting that siesta less than 40 min is associated with decreased risk of metabolic syndrome, while longer siesta is associated with increased risk. We aimed to investigate the effect of siesta duration on development of metabolic syndrome in a Mediterranean population using a prospective cohort study design. Methods: Our sample consisted of 9161 participants of the SUN cohort without components of metabolic syndrome at baseline. Siesta exposure was assessed at baseline and the development of metabolic syndrome components was assessed after an average 6.8 years of follow-up. We estimated odds ratios and fitted logistic regression models to adjust for potential cofounders including night-time sleep duration and quality, as well as other diet, health, and lifestyle factors. Results: We observed a positive association between average daily siesta >30 min and development of metabolic syndrome (aOR = 1.39 CI: 1.03–1.88). We found no significant difference in risk of developing metabolic syndrome between the group averaging ≤30 min of daily siesta and the group not taking siesta (aOR = 1.07 CI: 0.83–1.37). Further analysis suggested that average daily siesta <15 min may reduce risk of metabolic syndrome. Conclusions: Our study supports the J-curve model of the association between siesta and risk of metabolic syndrome, but suggests the protective effect is limited to a shorter range of siesta length than previously proposed.

## 1. Introduction

Siesta is a custom that involves lying down to rest for a few minutes to a couple of hours after having lunch. It is often perceived as a cultural practice of Mediterranean and Asian nations, but any review of the evidence reveals that it is far more widespread. Large studies estimate that more than 28% of the British populace take afternoon naps [1]—a figure which climbs above 40% in Switzerland, the Nordic countries, and the United States of America [2,3,4,5,6]. In this light, the siesta can be reconsidered a modern phenomenon, coinciding with the global reduction in night-time sleep hours and increased use of electronic devices, as well as the dominant neoliberal emphasis on economic productivity and trends that encourage working from home.

Studies investigating the health impacts of siesta often focus on neuropsychological outcomes: for example, whether a short siesta will improve post-nap mentation [7,8], or decrease sleepiness in drivers [9,10]. But there is also evidence that siesta affects the cardiovascular system. In 1987, Trichopolous et al. propounded the cardio-protective nature of siesta [11], however subsequent studies associated siesta with increased cardiovascular risk [12,13,14,15] and increased mortality [1,16,17,18,19,20,21,22]. This conflict, as well as the biological mechanisms responsible for it, remains unclear.

Metabolic syndrome is a clustering of cardiovascular risk factors: obesity, increased blood pressure, increased blood glucose, increased triglycerides, and lowered HDL. It is associated with increased risk of mortality [23,24], as well as type 2 diabetes mellitus [24,25,26], cancer [27,28], surgical complications [29,30], dementia [31,32], and heart failure [33]. Although metabolic syndrome is not considered a disease, research into its prevention is crucial for human health.

We aimed to use a prospective cohort study design to examine the effect of siesta on the development of metabolic syndrome. Most prior investigations into this relationship relied on cross-sectional data and were therefore unable to establish direction of causation. Our study follows a 2016 systematic review and meta-analysis, which found siesta under 40 min protective against metabolic syndrome and type 2 diabetes mellitus [34], as well as a 2013 prospective study in our own cohort, which found a 30 min siesta protective against obesity [35].

## 2. Methods

### 2.1. Study Design

The study is a prospective cohort study from the Seguimiento Universidad de Navarra (SUN) cohort. The SUN cohort is a large cohort comprising graduates from Spanish universities. It focusses on the study of diet and lifestyle, and related health outcomes. Recruitment began in 1999 and remains open indefinitely. Graduates are invited to participate upon leaving university, or through alumni and regional professional organizations. Over 50% of participants are health professionals. All data are self-reported. Upon entry into the cohort, participants complete a detailed baseline questionnaire, and shorter follow-up questionnaires are distributed every two years. Full details of the study methods have been published elsewhere [36].

### 2.2. Ethics

Ethics approval for the SUN cohort study was granted by the Institutional Review Board of the University of Navarra (2001_30). Submission of a completed baseline questionnaire is interpreted as evidence of free and informed consent, data are anonymized, and participants may withdraw from the study at any time.

### 2.3. Study Sample

We used the most current SUN database available at the time of analysis, dating from December 2019 and numbering 22,894 participants.

Figure 1 depicts the selection of the study sample. Participants were excluded from the sample if:-They did not have outcome data for any component of metabolic syndrome;-They met criteria for any component of metabolic syndrome at baseline;-Their baseline questionnaire responses did not meet minimum quality standards, as assessed by whether their Food Frequency Questionnaire yielded realistic values for energy intake [37];-They did not have baseline data about how long they slept at night.

Our selection criteria allowed that a participant was deemed not to meet obesity criteria for metabolic syndrome if intercurrent pregnancy affected the calculation of body mass index (BMI).

### 2.4. Exposure Assessment

Data on siesta habit were collected on the baseline questionnaire. Participants were asked about the average time they had spent doing various activities over the past year. Answer options ranged from “Never” through values from “<30 min per day” to “9+ hours per day”. Separate answers were provided for weekdays versus weekends. The data collection related to siesta has been validated in a subsample of the cohort using prospective records (intraclass correlation coefficients 0.68) [35].

We calculated average daily siesta time as a continuous variable, then transformed it into an ordinal variable with three categories: No Siesta, Short Siesta (less than or equal to 30 min), and Long Siesta (more than 30 min).

### 2.5. Outcome Assessment

Data related to components of metabolic syndrome were assessed on the third and fourth questionnaires, distributed approximately six and eight years after entry into the cohort. Metabolic syndrome and its components were defined according to the Harmonizing Definition [25]. Presence of each component was evaluated as a categorical yes/no variable, as was metabolic syndrome itself. Data were primarily drawn from the third questionnaire, however where measurements were missing, we attempted to recuperate them from the fourth questionnaire. Any data that remained missing after this process were assumed not to meet criteria for metabolic syndrome.

To assess metabolic syndrome components, waist circumference was defined in accordance with the International Diabetes Federation cut-offs for abdominal obesity in Europids: ≥80 cm in women and ≥94 cm in men [25]. Participants were provided with a tape measure and instructions for measuring waist and hip circumference. These self-measured and self-reported measurements for waist circumference have been validated in our cohort [38]. For any participant without waist circumference data, BMI ≥30 kg/m^2^ was used as criteria to determine obesity.

High blood glucose was defined as fasting glucose ≥100 mg/dL. Participants who reported a new diagnosis of diabetes, or who used insulin or oral anti-hyperglycaemic medications were also considered to fulfil criteria.

Participants met criteria for elevated blood pressure if they had systolic blood pressure of ≥130 mmHg, diastolic blood pressure of ≥85 mmHg, used anti-hypertensive medications, or had a current diagnosis of hypertension.

Participants met criteria for high triglycerides if they reported a triglyceride level ≥ 150 mg/dL, or if they took fibrate medication or fatty acid supplements.

Low HDL cholesterol was defined as HDL < 40 mg/dL for males, and HDL < 50 mg/dL for females.

### 2.6. Covariables

Information on relevant covariables and potential confounders was collected on the baseline questionnaire. Categorical covariables included sex, employment status (full-time/part-time/unemployed), special diets (yes/no), and medical diagnoses (yes/no for each of cardiovascular disease, cancer, depression and obstructive sleep apnea). Continuous covariables included length of tertiary study (years), working hours (hours/week), frequency of lunch at home (days), night-time sleep (hours), television viewing time (hours), smoking pack years (years), alcohol intake (g/day), social time (hours/day), and preceding weight gain (kg). Ordinal covariables included snoring (no/sometimes/yes), tendency to stress (range 0–10), year of entry into the cohort (1999–2014), and Mediterranean Diet Score (range 0–9) [39]. Covariables treated in quartiles included age (years), total daily energy intake (kcal), daily coffee intake (cups), and physical activity (METs/week). These covariables were treated in quartiles because, if treated as continuous variables, the wide distribution of responses made analysis so computationally intensive as to prohibit the inclusion of other covariables. Regression-based multiple imputation was used to impute missing values where required (working hours, lunch at home, television viewing, smoking pack years, BMI, social time, preceding weight gain, and tendency to stress).

### 2.7. Statistical Analysis

We estimated the crude odds ratios and associated 95% confidence interval (CI) for the development of metabolic syndrome using No Siesta as reference category. Next, we fitted logistic regression models to control for confounders and estimated adjusted odds ratios and 95% CI under two models: first adjusting for age and sex alone, then adjusting for the complete list of potential confounders. We also estimated the multivariate-adjusted odds ratio using short siesta as the reference category. We used logistic regression models to estimate multivariate-adjusted odds ratios for the development of each component of metabolic syndrome.

We assessed for effect modification by age, sex, sleep disorders (insomnia and obstructive sleep apnea), and adherence to or divergence from the recommended night-time sleep duration of 7–8 hours [40]. We used the test for likelihood ratio to estimate the *p* value for interaction.

For sensitivity analyses, we re-estimated multivariate-adjusted odds ratios using an alternate categorization of siesta exposure, focused on frequency of exposure, rather than daily average. These categories of siesta were: Never, Once or twice a week, Most days, Short Siesta once or twice a week, Short Siesta most days, Long Siesta once or twice a week, and Long Siesta most days.

All *p* values presented are two-tailed; *p* < 0.05 was considered statistically significant.

Statistical analysis was completed using STATA/MP 14.1.

## 3. Results

Out of 9161 participants included in our study (68.3% females, mean age 36.1 ± 10.5), siesta prevalence was 59.4% and 375 incident cases of metabolic syndrome (4.1%) were observed after a mean follow-up of 6.8 years.

Table 1 compares the baseline characteristics of participants across our total sample and in each category of siesta. Prevalence rates of cardiovascular disease, obstructive sleep apnea, and cancer were low (<2.5%) across the entire sample and in each group, although prevalence of cardiovascular disease and sleep apnea increased with longer siesta. Increased length of siesta also corresponded with increased prevalence of insomnia, increased prevalence of depression, increased energy and alcohol intake, increased smoking pack years, and more daily television. Compared to the total sample, the Long Siesta group had a higher proportion of men (only 66.93% were women as compared to 68.92% across the total sample) and reduced working hours (35.4 h/wk compared to 36.8 h/wk for the total sample).

Our odds ratio analysis for the development of metabolic syndrome according to length of siesta is presented in Table 2. We found a significant association between Long Siesta and the risk of metabolic syndrome when compared to reference category of No Siesta (multivariate-adjusted odds ratio OR = 1.39). The absolute risk for the development of metabolic syndrome was almost twice as high for Long Siesta (6.02%) compared to No Siesta (3.23%). There was no significant association between Long Siesta and metabolic syndrome when using Short Siesta as the reference category (OR = 1.30).

Table 3 presents the multivariate-adjusted odds ratios and 95% confidence intervals for development of each metabolic syndrome criteria. Obesity was the most frequently met criteria for metabolic syndrome (3662 cases; 40.0%): absolute risk increased from No Siesta (37.6%) to Short Siesta (40.9%), and Long Siesta had the highest absolute risk (43.4%). The multivariate-adjusted odds ratio for development of obesity under the condition of Long Siesta compared to No Siesta as reference category was 1.15 (95% CI 1.01–1.32). Taking No Siesta as reference, the absolute risks of developing any of high triglycerides, hypertension or hyperglycemia also increased with increasing siesta length, however in every case the CI for the adjusted OR included the null hypothesis. For low HDL, absolute risk was greatest under the condition of Long Siesta but was lowest for Short Siesta rather than for No Siesta. However, the findings for HDL were not statistically significant as the CI for the adjusted OR crossed the null hypothesis.

Figure 2 presents our exploration of non-linearity in the relationship between siesta exposure and development of metabolic syndrome. The positive relationship between siesta duration and development of metabolic syndrome is dose dependent and increases the longer the siesta continues beyond 30 min. However, for daily average siesta shorter than 15 min, the point value for the adjusted OR suggests a possible protective effect though the finding is not significant (aOR = 0.91, 95% CI 0.64–1.27).

Our stratified analysis (Table 4) identified a significant association between siesta and metabolic syndrome in participants who were younger than 50 years, participants without sleep disorders and participants whose night-time sleep duration was less than 7 h or more than 8 h. No association between siesta and metabolic syndrome was found in stratified analysis for either men or women. We found no evidence of statistical interaction between age, sex, or sleep disorder. The results of our sensitivity analyses are presented in Appendix A.

## 4. Discussion

### 4.1. Interpretation of Findings Related to Metabolic Syndrome

In this Mediterranean population of Spanish university graduates, we found a positive association between daily siesta of more than 30 min duration and the development of metabolic syndrome.

Our findings concur with previous studies into siesta and metabolic syndrome which have found that siesta of long duration increases the risk of metabolic syndrome, while siesta of shorter duration has indeterminate or even protective effect [3,34,41,42,43,44]. In a 2016 review and meta-analysis incorporating 288,883 participants, Yamada et al. demonstrated that the evidence points toward a J-curve relationship [34] and more recent studies also corroborate this suggestion [41]. Our study is important because it offers longitudinal evidence whereas all other contributing studies have been cross-sectional.

Beyond the general shape of the curve, agreement between previous studies has been limited. There is no convergence of opinion regarding what length of siesta is most protective, or at what length of siesta the association with metabolic syndrome becomes positive. In their review article, Yamada et al. observed a protective effect of siesta shorter than 40 min, and can only be confident of a positive association between siesta and metabolic syndrome when duration exceeds 60 min daily. By contrast, in a more recent cross-sectional study, He et al. suggest that naps exceeding 90 min increase risk of metabolic syndrome, while naps of approximately 30 min may protect against it. Different categorization of siesta length complicates comparison between studies.

### 4.2. Interpretation of Findings Related to Metabolic Syndrome Components

#### 4.2.1. Obesity

We found that average daily siesta of more than 30 min duration was positively associated with obesity (aOR 1.15, 95% CI 1.01–1.32). Several cross-sectional studies have similarly associated longer siesta with obesity [42,44,45,46], however one Chinese cross-sectional study into on a related outcome associated a 0.1–1 hr nap habit with reduced risk of being overweight as determined by BMI ≥ 25 kg/m^2^ [47] and a previous prospective study from our own SUN cohort found that daily siesta of 30 min was associated with decreased risk of obesity (HR 0.67; 95% CI 0.46–0.96) [35]. This apparent difference between two studies from the same cohort stems from the different methods of measuring obesity (the previous study using BMI rather than waist circumference); different sampled populations (the previous study excluded only participants with obesity at baseline, whereas we excluded participants with any metabolic syndrome criteria); and different categorization of siesta (the prior study had separate groups for siesta equal to 30 min and siesta greater than 30 min, but in our study these two categories were combined).

#### 4.2.2. Triglycerides

We found that daily siesta averaging more than 30 min duration increased the risk of high triglycerides (aOR 1.33, 95% CI 1.00–1.76). Positive associations between longer and more frequent siesta and high triglycerides have been previously described in cross-sectional studies [41,42,48].

#### 4.2.3. Remaining Metabolic Syndrome Criteria

We did not find any relationship between siesta length and blood pressure or siesta length and elevated blood glucose, despite previous studies demonstrating positive associations between longer siesta and type 2 diabetes mellitus [49,50] and between siesta over 30 min and hypertension [51]. In our study, point estimates for the relationship between siesta over 30 min and these outcomes were positive, but the 95% confidence intervals crossed the value for null effect and the *p* value was high. Similarly, our data for LDH cholesterol did not demonstrate any significant relationship.

### 4.3. Limitations

Our findings are limited by the complexity of defining a variable to analyze siesta exposure. We defined our exposure variable as the average daily siesta duration across a week, however in doing so we blurred the possibly pertinent distinction between duration of a single siesta and frequency of siesta practice across the week. In sensitivity analysis we created alternate variables that focused on frequency of siesta length, thereby preserving the distinction between these two characteristics of exposure. These sensitivity analyses confirmed our main finding that daily siesta of greater than 30 min increases risk of metabolic syndrome. Furthermore, it supported a suggestion that siesta of less than or equal to 30 min duration seems to confer some protection against metabolic syndrome, while siesta of over 30 min does not.

Our categorization of siesta was also limited by our inability to guarantee stable siesta habits over time. Baseline assessment of siesta occurred at the mean age of 35, Furthermore, the chances that an individual would be able to maintain a consistent siesta habit over time is increased in the Spanish context because the siesta is a socially integrated practice. Nonetheless, changing employment, changing seniority within the same employment (especially in the health care setting), and changing stages of life are likely to affect a person’s ability to practice consistent siesta, and the resultant categorisation error could strongly influence our results. The effect of this error would most likely would push our results toward the null hypothesis, however it could also introduce a new variable that we have not accounted for. In practice, our age stratified analysis found that, compared to No Siesta, Long Siesta significantly increased risk of metabolic syndrome for participants aged <50 at baseline, those whose siesta habit would be most likely to be changeable. 

Another potential limitation of our study is our reliance on self-reported data. However, the data relating to siesta has been previously validated using prospective records and was found to have an acceptable intraclass correlation coefficient [35]. In addition, self-reported measurements of metabolic syndrome components and diagnosis of metabolic syndrome have been validated in this cohort, yielding sensitivity of 66% and specificity of 98% for diagnosis of metabolic syndrome and good validity for research with positive and negative predicative values of over 90% [38,52]. In any case, the error potentially introduced by self-reporting would be expected to be non-differential as the bias introduced would tend toward the null hypothesis. As such, it should not undermine our positive findings.

Most importantly, we were restricted in our ability to analyze different lengths of siesta less than 30 min as our data collection questionnaire did not provide sufficient detail to separate different lengths of siesta under 30 min.

### 4.4. Strengths

The strengths of the present study include its dynamic participation, prospective design, long follow-up period, and high retention rate. We were also able to adjust for many relevant covariables, including those related to night-time sleep duration and quality—which other studies have been unable to adjust for, as well as important aspects of diet and lifestyle, and relevant other health conditions. Further, our cohort is drawn from a highly educated sub-section of the Spanish population, which may further help us to minimize confounding related to socio-economic differences. Our ability to minimize the effect of confounders improves the generalizability of our results so that we may cautiously apply them to other Mediterranean adults free from metabolic syndrome components.

### 4.5. Biological Explanation

It has often been suggested that siesta is associated with cardiovascular mortality due to ongoing harm exerted by the rise in blood pressure and the cardiometabolic changes that occur upon waking [12,14,16]. This theory suggests that taking siesta increases the number of times per day that one wakes up from sleep, and thereby increases related cardiovascular risk [53].

Yet this theory does not explain the relationships we observed. We did not find any significant association between siesta and elevated blood pressure, and we were not studying the outcome of cardiovascular mortality. Perhaps this often-cited theory may be most relevant in people who already have hypertension or cardiovascular disease at the time of exposure to siesta, and for this reason did not apply to our sample which was free from such diseases at baseline exposure.

It is our opinion that a large part of the biological explanation for the observed association between siesta and metabolic syndrome will be found in the study of circadian rhythm [54,55,56]. Circadian rhythmicity strongly affects sleep timing and quality as well as energy intake, energy expenditure, and physiological responses to food. Circadian disruption has been found to impair glucose tolerance, reduce energy expenditure, and increase appetite and energy intake [55,56,57,58]. Circadian disruption—in the form of social jetlag, shift work, and exposure to light at night—has also been associated with metabolic syndrome, weight gain, and cardiovascular mortality [55,56,58,59]. We propose that siesta may disrupt circadian rhythmicity either through daytime exposure to darkness [55] or if its duration exceeds the duration of the dip in cortisol that is believed to occur physiologically in the early afternoon.

### 4.6. Conclusions

Media coverage often reports the benefits of siesta in improving cognition, alertness, mood, and productivity. But the public is less likely to be aware of studies linking siesta with cardiovascular risk and obesity. As greater numbers of people work from home, partial or unbalanced representations of the impacts of siesta could encourage more people to take siesta, under the impression it may benefit their health.

As such, it is crucial to share evidence that long siesta can increase the risk of obesity and cardiovascular disease, even in healthy individuals. The public would benefit from advice that siesta should be taken occasionally and limited to less than 30 min. They should also be made aware of other ways to boost afternoon alertness and cognition, such as exercise or brisk walking, and health professionals should emphasize the benefits of getting enough sleep at night. For those experiencing sleep disorders, methods other than siesta should be recommended to minimize negative impacts of sleep deficit.

However, given that advice to boost productivity often recommends short naps of 10–20 min, it seems possible that cognitive benefits and cardiovascular concerns could be balanced. More research needs to be done to identify what duration of siesta could reduce cardiac risk factors while benefiting cognitive function.

## Figures and Tables

**Figure 1 nutrients-13-04182-f001:**
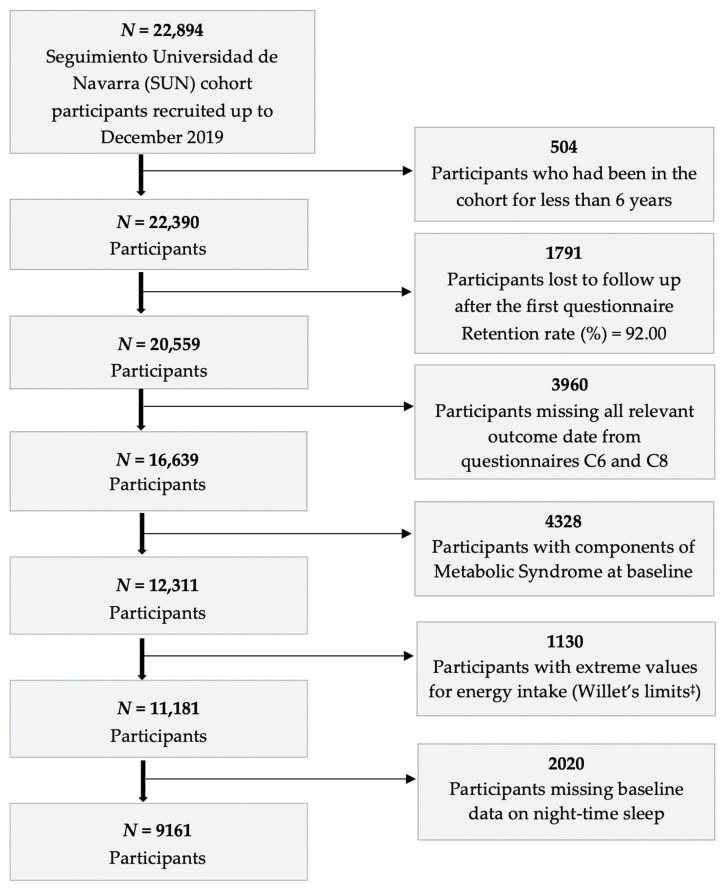
Flowchart depicting the selection process among participants of the SUN project (Seguimiento Universidad de Navarra, University of Navarra Follow-up) included in the present analyses. Navarra, Spain, 1999–2019. ^‡^ Willet 2012.

**Figure 2 nutrients-13-04182-f002:**
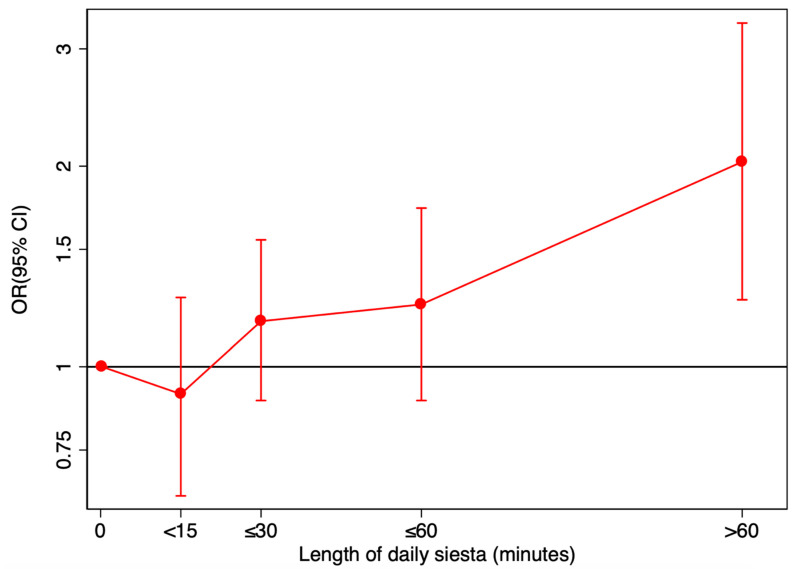
A non-linear relationship: Adjusted odd ratios (aOR) * and 95% confidence intervals (CI) for the development of metabolic syndrome according to length of daily siesta in the SUN cohort. * Adjusted for sex, age, years of university, year of entry into cohort, working hours, lunch at home, hours of night-time sleep, daily TV, smoking pack-years, daily alcohol intake, total daily energy intake, daily coffee intake, Mediterranean Diet Score, special diets, physical activity, social time, prevalent cardiovascular disease, prevalent cancer, prevalent depression, prior history of insomnia, obstructive sleep apnea, snoring, weight gain prior to recruitment, tendency to stress.

**Table 1 nutrients-13-04182-t001:** Baseline characteristics of sample participants.

Characteristic	Total Sample	No Siesta (0 min/Day)	Short Siesta (≤30 min/Day)	Long Siesta (>30 min/Day)	*p*
*N*	9161	3719	3897	1545	
Women (%)	68.92%	69.86%	67.33%	66.93%	0.03
Age (years) (M ± SD)	36.1 ± 10.5	35.3 ± 10.3	36.9 ± 10.1	36.1 ± 11.6	<0.001
Year of entry into cohort (M ± SD)	2003 ± 2.9	2002 ± 2.9	2003 ± 3.0	2003 ± 2.8	<0.001
Years of tertiary study (years) (M ± SD)	5.1 ± 1.5	5.1 ± 1.5	5.1 ± 1.5	4.9 ± 1.4	<0.001
Employment status:					<0.001
Full-time (%)	72.76%	68.73%	77.34%	70.94%	
Part-time (%)	9.26%	10.11%	8.83%	8.28%	
Other (%)	17.98%	21.16%	13.83%	20.78%	
Working hours (h/wk) (M ± SD)	36.8 ± 16.7	36.2 ± 17.5	38.0 ± 15.6	35.4 ± 17.2	<0.001
Lunch at home (days/wk) (M ± SD)	5.5 ± 2.1	5.4 ± 2.1	5.4 ± 2.1	5.9 ± 1.7	<0.001
Night-time sleep (h/night) (M ± SD)	7.4 ± 0.9	7.4 ± 0.9	7.4 ± 0.8	7.3 ± 1.1	<0.001
Daily television (h/day) (M ± SD)	1.6 ± 1.3	1.5 ± 1.4	1.5 ± 1.0	2.0 ± 1.5	<0.001
Smoking pack-years (pack-year) (M ± SD)	4.6 ± 7.8	3.8 ± 7.0	4.8 ± 7.8	5.7 ± 9.4	<0.001
Alcohol (g/day) (M ± SD)	5.8 ± 8.3	4.9 ± 7.5	6.3 ± 8.3	6.7 ± 9.8	<0.001
Total energy intake (kcal/day) (M ± SD)	2362.2 ± 594.4	2351.6 ± 592.6	2362.0 ± 590.4	2388.3 ± 608.3	0.13
Coffee intake (cups/day) (M ± SD)	1.2 ± 1.2	1.3 ± 1.3	1.3 ± 1.2	1.1 ± 1.3	<0.001
Mediterranean Diet Score (score out of 9) (M ± SD)	4.2 ± 1.8	4.0 ± 1.8	4.3 ± 1.8	4.3 ± 1.8	<0.001
Special diets (%)	6.27%	6.16%	6.21%	6.67%	0.77
Physical activity (METs-h/week) (M ± SD)	20.9 ± 22.4	20.7 ± 22.5	21.0 ± 21.8	21.2 ± 23.5	0.75
Social time (h/day) (M ± SD)	0.6 ± 0.4	0.5 ± 0.4	0.5 ± 0.4	0.6 ± 0.4	<0.001
Prevalent cardiovascular disease (%)	0.67%	0.59%	0.69%	0.78%	0.73
Prevalent cancer (%)	2.16%	2.02%	2.36%	2.01%	0.53
Prevalent depression or use of antidepressant medication (%)	10.53%	9.49%	10.37%	13.46%	<0.001
Previous history of insomnia (%)	16.88%	15.46%	17.35%	19.09%	<0.001
Obstructive sleep apnea (%)	0.75%	0.67%	0.80%	0.84%	0.75
Snoring (%)	13.66%	11.86%	14.32%	16.31%	<0.001
Weight gain prior to C0 (kg) (M ± SD)	0.8 ± 4.0	0.7 ± 4.0	0.8 ± 3.9	1.0 ± 4.1	0.01
Tendency to stress (0–10) (M ± SD)	6.0 ± 2.2	6.0 ± 2.2	6.0 ± 2.2	6.0 ± 2.2	0.23

Continuous variables are expressed as mean with standard deviation in parentheses and categorical variables are expressed as percentages.

**Table 2 nutrients-13-04182-t002:** Adjusted odds ratios (aOR) * and 95% confidence intervals (CI) for the development of metabolic syndrome according to siesta length in the SUN cohort.

	Siesta Length		
	No Siesta	Short Siesta (≤30 min)	Long Siesta (>30 min)
*N*	3719	3897	1545
Cases	120	162	93
% Absolute risk (cases/*N*)	3.23%	4.16%	6.02%
Crude OR (95% CI)	1 Ref.	1.30 (1.02–1.65)	1.92 (1.46–2.54)
Age and Sex adjusted OR (95% CI)	1 Ref.	1.15 (0.89–1.47)	1.59 (1.19–2.13)
Multivariable adjusted OR (95% CI)	1 Ref.	1.07 (0.83–1.37)	1.39 (1.03–1.88)
Crude OR (95% CI)	0.77 (0.60–0.98)	1 Ref.	1.48 (1.14–1.92)
Age and Sex adjusted OR (95% CI)	0.87 (0.68–1.12)	1 Ref.	1.39 (1.05–1.83)
Multivariable adjusted OR (95% CI)	0.94 (0.73–1.21)	1 Ref.	1.30 (0.98–1.73)

* Adjusted for sex, age, years of university, year of entry into cohort, working hours, lunch at home, hours of night-time sleep, daily TV, smoking pack-years, daily alcohol intake, total daily energy intake, daily coffee intake, Mediterranean Diet Score, special diets, physical activity, social time, prevalent cardiovascular disease, prevalent cancer, prevalent depression, prior history of insomnia, obstructive sleep apnea, snoring, weight gain prior to recruitment, tendency to stress.

**Table 3 nutrients-13-04182-t003:** Adjusted odd ratios (aOR) * and 95% confidence intervals (CI) for the development of metabolic syndrome criteria according to siesta length in the SUN cohort.

	Siesta Length		
Metabolic Syndrome Criteria	No Siesta	Short Siesta (≤30 min)	Long Siesta (>30 min)
Obesity (Waist Circumference ≥ 80 cm in women or ≥ 94 cm in men or BMI ≥ 30 kg/m^2^)			
*N* ** = 9161	*N* = 3719	*N* = 3897	*N* = 1545
Cases = 3662	1399	1593	670
% Absolute risk = 39.97%	37.62%	40.88%	43.37%
Multivariable adjusted OR (95% CI)	1 Ref.	1.04 (0.94–1.15)	1.15 (1.01–1.32)
High triglycerides (Serum triglycerides ≥ 150 mg/dL or pharmacological treatment for high triglycerides)			
*N* ** = 5345	*N* = 2149	*N* = 2294	*N* = 902
Cases = 417	142	185	90
% Absolute risk = 7.80%	6.61%	8.06%	9.98%
Multivariable adjusted OR (95% CI)	1 Ref.	1.10 (0.88–1.39)	1.33 (1.00–1.76)
Low HDL cholesterol (Serum HDL cholesterol < 50 mg/dL in women or <40 mg/dL in men)			
*N* ** = 4705	*N* = 1856	*N* = 2054	*N* = 795
Cases = 488	199	193	96
% Absolute risk = 10.37%	10.72%	9.39%	12.08%
Multivariable adjusted OR (95% CI)	1 Ref.	0.90 (0.73–1.10)	1.13 (0.87–1.47)
Hypertension (Systolic blood pressure ≥ 130 mmHg or diastolic blood pressure ≥ 85 mmHg or pharmacological treatment for hypertension)			
*N* ** = 7724	*N* = 3079	*N* = 3316	*N* = 1329
Cases = 1486	550	654	282
% Absolute risk = 19.24%	17.86%	19.72%	21.22%
Multivariable adjusted OR (95% CI)	1 Ref.	1.01 (0.88–1.15)	1.09 (0.92–1.30)
Hyperglycaemia (Fasting glucose ≥ 100 mg/dL or pharmacological treatment for hyperglycaemia)			
*N* ** = 6616	*N* = 2615	*N* = 2855	*N* = 1146
Cases = 759	266	344	149
% Absolute risk = 11.47%	10.17%	12.05%	13.00%
Multivariable adjusted OR (95% CI)	1 Ref.	1.09 (0.92–1.30)	1.10 (0.88–1.38)

* Adjusted for sex, age, years of university, year of entry into cohort, working hours, lunch at home, hours of night-time sleep, daily TV, smoking pack-years, daily alcohol intake, total daily energy intake, daily coffee intake, Mediterranean Diet Score, special diets, physical activity, social time, prevalent cardiovascular disease, prevalent cancer, prevalent depression, prior history of insomnia, obstructive sleep apnea, snoring, weight gain prior to recruitment, tendency to stress. ** For each component, the number of participants is restricted to those with outcome data available for the specified component. Follow up time for each component of metabolic syndrome is as follows: 6.6 years for obesity, 6.8 years for fasting glucose, 6.7 years for blood pressure, 6.9 years for triglycerides and 6.9 years for HDL cholesterol.

**Table 4 nutrients-13-04182-t004:** Analysis of effect modification: Adjusted odd ratios (aOR) * and 95% confidence intervals (CI) for the development of the metabolic syndrome according to siesta length in the SUN cohort stratified by potential confounders.

	Siesta Length			
Potential Modifier of Effect of Siesta	No Siesta	Short Siesta (≤30 min)	Long Siesta (>30 min)	*p* for Interaction
	*N* = 3719	*N* = 3897	*N* = 1545	
Age				0.354
Age < 50 years (*N* = 8080)				
*N*	3339	3422	1319	
Cases	71	104	49	
aOR (95% CI)	1 Ref.	1.18 (0.86–1.61)	1.53 (1.04–2.27)	
Age ≥ 50 years (*N* = 1081)				
*N*	380	475	226	
Cases	49	58	44	
aOR (95% CI)	1 Ref.	0.87 (0.56–1.33)	1.31 (0.81–2.12)	
Sex				0.368
Men (*N* = 2905)				
*N*	1121	1273	511	
Cases	66	87	57	
aOR (95% CI)	1 Ref.	0.94 (0.66–1.33)	1.35 (0.90–2.03)	
Women (*N* = 6256)				
*N*	2598	2624	1034	
Cases	54	75	36	
aOR (95% CI)	1 Ref.	1.20 (0.83–1.73)	1.45 (0.92–2.29)	
Sleep disorder				0.879
No sleep disorder (*N* = 7554)				
*N*	3118	3199	1237	
Cases	94	124	68	
aOR (95% CI)	1 Ref.	1.10 (0.83–1.47)	1.42 (1.01–2.01)	
Sleep disorder (*N* = 1607)				
*N*	601	698	308	
Cases	26	38	25	
aOR (95% CI)	1 Ref.	1.00 (0.58–1.71)	1.27 (0.68–2.38)	
Night-time sleep duration				0.104
7–8 h of nightly sleep (*N* = 6358)				
*N*	2622	2837	899	
Cases	85	117	47	
aOR (95% CI)	1 Ref.	1.08 (0.80–1.46)	1.19 (0.80–1.76)	
<7 or >8 h of nightly sleep (*N* = 2803)				
*N*	1097	1060	646	
Cases	35	45	46	
aOR (95% CI)	1 Ref.	1.06 (0.66–1.71)	1.71 (1.04–2.80)	

* Adjusted for sex, age, years of university, year of entry into cohort, working hours, lunch at home, hours of night-time sleep, daily TV, smoking pack-years, daily alcohol intake, total daily energy intake, daily coffee intake, Mediterranean Diet Score, special diets, physical activity, social time, prevalent cardiovascular disease, prevalent cancer, prevalent depression, prior history of insomnia, obstructive sleep apnea, snoring, weight gain prior to recruitment, tendency to stress.

## Data Availability

Available upon request to Department of Preventive Medicine and Public Health, School of Medicine, University of Navarra, Pamplona, Spain.

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
