# Peer review of "Risk of Developing Metabolic Syndrome Is Affected by Length of Daily Siesta: Results from a Prospective Cohort Study"

_nutrients, 2021, doi:10.3390/nu13114182_

Round 1
Reviewer 1 Report
- A siesta should be defined, is it a nap or only a rest in a sitting position or maybe some physical exercise such as a walk.
- In introduction line 59 “lowered LDL” if HDL?
- Liter with capital letter L not l.
- A lot of authors use abbreviation MetS for metabolic syndrome, could be considered here.
- Lines 92-95 should be moved to left.
- In the Statistical analysis “estimated” should be used instead of “calculate”, and p is not a trend.
- All tables and the description of the results should be corrected. All tables should be extended for entire page without left margin.
- Change font into Pallatino linotype.
- In table 1 add column 2 with Total, add the last column with p to compare characteristics between 3 groups: no siesta, short siesta and long siesta. In every row add M±SD for numerical variables or n (%) for categorical variables and change results presentation according to those remarks and correct lines 181-187 according to that.
- Lines 191-196 – write reference to Table 2, line 193 parentheses for OR shoulb be deleted, p=0.03 not trend, CI is not needed, line 194 – “6.0%” should be presented in table 2, and 3.2% where is presented and how estimated – add % with cases/N. are p’s for short siesta or long siesta compared to no siesta? Line 196 OR=1.30 , 95% CI: 0.98-1.73 – in table 2 different result is presented. So add results for log siesta with reference short siesta in Table 2, p not for trend in Table 2.
- Line 206 36662 cases add %. Line 209 versus what? No siesta? There are no descriptions for other MetS component.
- Table 3 add %, p not for trend, p for long siesta or for short siesta?
- Page 7, table, line 6, diastostic twice, one should be deleted.
- Line 225 – not “relationship is inverted” but no relationship.
- Line 226 – OR not significant.
- Figure 2, axle scales are not correct, not proportional, add labels for all OR and p, at least one p is not significant for <15.
- Line 236 add gender.
- Line 238 “We found no evidence of statistical interaction” between what variable and what variable affect what variable.
- Line 264-265 – “are positively associated with…” should be “increased risk…
- Line 265 “ are protective” against what?
- Line 284 “with both high triglycerides” and what?
- Supplementary table 1 – line 1 row 1 no heading- should be added :frequency of siesta habit”.
- Why the authors presents so many characteristics in table 1 – which of them are correlated with siesta duration or MetS prevalence?
Author Response
Reply to Reviewer 1:
Thank you so much for your detailed and constructive feedback. We sincerely appreciate the time and effort that you have put in so that we might improve our article. Your advice has been heeded and we have integrated your suggestions into our work. The detailed point-by-point breakdown is as follows, with your suggestions listed in black and the changes we have made outlined in blue.
Reviewer 1 Comments
- A siesta should be defined, is it a nap or only a rest in a sitting position or maybe some physical exercise such as a walk.
We have made the suggested change. We have defined siesta as “a custom consisting of lying down to rest for a few minutes to a couple of hours after having lunch.” Line 43
- In introduction line 59 “lowered LDL” if HDL?
We have made the suggested change. We have corrected line 59 to “HDL”
- Liter with capital letter L not l.
We have made the suggested change.We have corrected the SI units for millilitre and decilitre to mL and dL
- A lot of authors use abbreviation MetS for metabolic syndrome, could be considered here.
Thank you for the suggestion. For the time being, we have chosen to retain “metabolic syndrome” rather than “MetS” as per the advice of reviewer 2.
- Lines 92-95 should be moved to left.
We have made the suggested change. We have corrected the alignment of lines 92-95
- In the Statistical analysis “estimated” should be used instead of “calculate”, and p is not a trend.
We have made the suggested change. In the description of Statistical Analysis we have replaced “calculated” by “estimated”, and we have removed incorrect mention of “p for trend”. The p for comparison between each of the categories is found in the 95% CI.
- All tables and the description of the results should be corrected. All tables should be extended for entire page without left margin.
We have made the suggested change. We have corrected the alignment of tables and description of results
- Change font into Pallatino linotype.
We have made the suggested change. We have corrected the font to Palatino Linotype for all text and tables
- In table 1 add column 2 with Total, add the last column with p to compare characteristics between 3 groups: no siesta, short siesta and long siesta. In every row add M±SD for numerical variables or n (%) for categorical variables and change results presentation according to those remarks and correct lines 181-187 according to that.
We have made the suggested changes.
- We have added column 2 with Total
- We have added the last column with p to compare characteristics between the three groups
- We have added (M±SD) to numerical variables and (%) to all categorical variables listed in column 1. We have corrected the results presentation to include % as well as M±SD. We have corrected lines 181-197 in accordance.
- Lines 191-196 – write reference to Table 2, line 193 parentheses for OR shoulb be deleted, p=0.03 not trend, CI is not needed, line 194 – “6.0%” should be presented in table 2, and 3.2% where is presented and how estimated – add % with cases/N. are p’s for short siesta or long siesta compared to no siesta? Line 196 OR=1.30 , 95% CI: 0.98-1.73 – in table 2 different result is presented. So add results for log siesta with reference short siesta in Table 2, p not for trend in Table 2.
We have made the suggested changes. In the paragraph preceding Table 2 (lines 191-196 of original document) and in Table 2 itself,
- We have added a reference to Table 2
- We have removed parentheses for OR
- We have removed P for trend
- We have included % Absolute risk with cases/N in Table 2
- We had added results with Short Siesta as reference category
- Line 206 36662 cases add %. Line 209 versus what? No siesta? There are no descriptions for other MetS component.
We have made the suggested changes to this paragraph,
- We have added the absolute risk (%) for 3662 cases
- We have clarified the comparison in line 209 (Reference Category No Siesta)
- We have added in text descriptions for the findings relating to the other metabolic syndrome criteria
- Table 3 add %, p not for trend, p for long siesta or for short siesta?
We have made the suggested changes. In Table 3,
- We have added % Absolute risk (cases/N)
- We have removed mention of “p for trend”. The p for comparison between categories is found in the 95% CI.
- Page 7, table, line 6, diastostic twice, one should be deleted.
We have made the suggested change. We have removed the repeated word.
- Line 225 – not “relationship is inverted” but no relationship.
We have made the suggested change. We have corrected our comment “the relationship is inverted” to convey that there is no relationship: “the point value of the adjusted OR suggests a possible protective effect although the CI is not significant”
- Line 226 – OR not significant.
As above
- Figure 2, axle scales are not correct, not proportional, add labels for all OR and p, at least one p is not significant for <15.
We have made the suggested changes. We have corrected the scale for the x axis so that it is proportional. The values for CI are expressed in the error bars. P is not significant in most categories, except for larger values of x.
- Line 236 add gender.
We have made the suggested change. In the paragraph before Table 4, we have added a comment about sex as potential modifier
- Line 238 “We found no evidence of statistical interaction” between what variable and what variable affect what variable.
We have made the suggested change. We have added detail regarding the variables examined for statistical interaction
- Line 264-265 – “are positively associated with…” should be “increased risk…
We have made the suggested change. We have replaced the phrase “are positively associated with” to “increased risk of”
- Line 265 “ are protective” against what?
We have made the suggested change. We have clarified “protective against [metabolic syndrome]”
- Line 284 “with both high triglycerides” and what?
We have corrected our mistake and removed the word “both”
- Supplementary table 1 – line 1 row 1 no heading- should be added :frequency of siesta habit”.
We have made the suggested change. In supplementary table 1, we have added the heading “Frequency of siesta habit”
- Why the authors presents so many characteristics in table 1 – which of them are correlated with siesta duration or MetS prevalence?
Thank you for challenging us to defend the number of variables included in Table 1. We are aware that there are many variables, however there is a reason for their inclusion. Table 1 includes all variables that we adjusted for in the logistic regression analysis as well as all variables that were plausibly EITHER a) Independently associated with the outcome (Metabolic Syndrome) without being a result of an effect of Siesta OR b) Associated with the exposure to siesta itself, independently of its effect AND were c) not intermediary between the exposure and the effect.
Reviewer 2 Report
It seems that there is no need to use capital letters for metabolic syndrome
v.128: the units of BMI should be added
It would be worth mentioning in the „Discussion” the problem of possibly different length of siesta in the young graduates and during the following years; one could guess that sjesta in students and sjesta in working people (especially in health services) might not be of the same or even similar duration. This could strongly influence the results.
Author Response
Reply to Reviewer 2
Thank you very much for your positive assessment of our work and for your helpful comments. Based in your advice, we have made the following improvements to our article (your suggestions are listed in black, our responses in blue):
Reviewer’s comments
- It seems that there is no need to use capital letters for metabolic syndrome
We have made the suggested change. We have removed capital letters for metabolic syndrome.
- 128: the units of BMI should be added
We have made the suggested change. We have added units for BMI.
- It would be worth mentioning in the „Discussion” the problem of possibly different length of siesta in the young graduates and during the following years; one could guess that siesta in students and siesta in working people (especially in health services) might not be of the same or even similar duration. This could strongly influence the results.
Thank you for this comment. We have added a paragraph to our discussion that sets forth the limitation you correctly identify and to consider its important effect.